# Sleep quality and associated factors among people with asthma at public hospitals in east gojjam zone, North West Ethiopia, 2022

Menberu Gete[1]*, Mezinew Sintayehu Bitew[1], Tirusew Wondie[1], Bekalu Bewket[2,7], Haile Amiha[1], Henok Mulugeta[1,3], Wuhabie Tsega Sahilu[1], Balew Adane[4], Aster Tadesse[5], Baye Tsegaye Amlak[1], Dejen Tsegaye Alem[1,6], Tiliksew Liknaw Alemneh[1], Asmamaw Getnet[1]

1 Department of Nursing, College of Health Sciences, Debre Markos University, Debre Markos, Ethiopia, 2 Federal State of Parana University, Brazil, 3 Center for Chronic & Complex Care Research, School of Nursing, University of Wollongong Blacktown Hospital, Western Sydney Local Health District, 4 Department of Environmental Health, College of Health Science, Debre Markos University, Debre Markos, Ethiopia, 5 Department of Pediatrics and Child Health Nursing, College of Health Science, Debre Markos University, Debre Markos, Ethiopia, 6 Graduate School of Medicine, Faculty of Science, Medicine and Health, University of Wollongong, Australia 7 Department of Nursing, College of Health Science, Injibara University, Injibara, Ethiopia.

* menberu_gete@dmu.edu.et

## Abstract

### Introduction

The magnitude of poor sleep quality among people with asthma is widespread and has detrimental consequences, including a higher chance of having poor work performance, an increase in the frequency of asthma attacks, an increase in the need for overnight hospitalization, and a worse health related quality of life. However, it has not been well studied, especially in low-income countries like Ethiopia. This study's objective was to assess the degree of sleep quality and related factors among people with asthma who had follow-up visits at public hospitals in the East Gojjam Zone.

### Methods

An institutional-based cross-sectional study design was conducted among 406 people with asthma through consecutive sampling techniques at public hospitals in East Gojjam Zone from June 6 to July 1, 2022. Sleep quality was measured by the Pittsburgh Sleep Quality Index through a face-to-face interview, and the collected data were entered into Epi Data version 4.4.2 and exported to SPSS version 25 for analysis. Logistic regression was fitted to assess the association between dependent and independent variables. Variables with a P-value <0.05 in multivariable logistic regression were considered statistically significant for the dependent variable.

**Data availability statement:** All relevant data are within the paper and its Supporting Information files.

**Funding:** The author(s) received no specific funding for this work.

**Competing interests:** The authors have declared that no competing interests exist

**Acronyms and Abbreviations:** ACT–Asthma Control TestAOR–Adjusted Odd RatioASSIST–Alcohol Smoking and Substance Involvement Screening TestBPH–Bichena Primary HospitalBIPH–Bibugn Primary HospitalCI–Confidence IntervalCOR–Crude Odd RatioDALYs–Disability-Adjusted Life-YearsDEPH–Debre Elias Primary HospitalDMCSH–Debre Markos Comprehensive Specialized HospitalDPH–Dejen Primary HospitalDWPH–Debre Work Primary HospitalsGINA–Global Initiative for AsthmaICS–Inhaled CorticosteroidsLPH–Lumamie Primary HospitalMGH–Motta General HospitalPSQI–Pittsburgh Sleep Quality IndexSBPH–Shebel Berenta Primary HospitalSDI–Socio-Demographic IndexSOB–Shortness of BreathSPSS–Statistical Project Service SolutionSSD–Short Sleep DurationWHO–World Health OrganizationYLDs–Years Lived With DisabilityYLLs–Years of Life LostYPH–Yejubie Primary Hospital

## Results

Among participants, 64.5% had poor sleep quality. Having anxiety (AOR = 2.67), comorbidity (AOR = 3.11), poor sleep hygiene practices (AOR = 2.99), severe asthma (AOR = 2.31), very poorly controlled asthma (AOR = 4.00) and current alcohol use (AOR = 3.20) were associated with poor sleep quality.

## Conclusion

The magnitude of poor sleep quality was high in this study. Severe asthma, having comorbidity, very poorly controlled asthma, poor sleep hygiene practices, having anxiety symptoms and current use of alcohol were variables associated with poor sleep quality. Reduction in alcohol consumption, improving sleep hygiene practices, and treating comorbidities would be effective measures for improving poor sleep quality. Individuals with severe asthma and uncontrolled asthma symptoms need special attention.

## Introduction

Asthma is a chronic lung disease characterized by airway hyperresponsiveness to various direct and indirect stimuli, leading to variable symptoms of wheeze, and shortness of breath (SOB), chest tightness, cough, and variable expiratory airflow limitation [1]. Globally, It affects an estimated 262 million (15.7%) people and is responsible for 461000 (5.4%) deaths and 21·6 million (5.6%) disability-adjusted life-years (DALYs), 10.2 million (15.4%) years lived with disability (YLDs) and 11.4 million (1.9%) years of life lost (YLLs) in 2019 [2]. In Africa, it affects an estimated 50,668,000 (4.2%) people and causes 62,544 (0.7%) deaths [3].

Sleep is a crucial human physiological process important for the proper functioning of the brain and the overall well-being of the human body. It accounts for approximately an individual's life [4].

Sleep quality is an individual's self-satisfaction with all aspects of the sleep experience [5].

Sleep disturbance refers to persistent difficulties initiating sleep, frequent awakenings and inability to return to sleep [6]. In individuals with asthma, these disturbances may be due to various physiological mechanisms, including increased bronchial hyper-responsiveness and changes in inflammatory path-ways [7].

Many hormonal changes, including epinephrine, cortisol, and melatonin, that occur in the evening may contribute to inflammation in the airways, which causes airways to narrow, increasing the risk of nocturnal asthma symptoms and leading to sleep problems[8]

Poor sleep quality is very common and worthy of concern among people with asthma [9]. Its prevalence is high in both high and low income countries ranging from 54.5% to 80.49% [9–17].

People with asthma who have reduced sleep quality are more likely to have a significantly reduced overall and disease-specific quality of life [14,18]. They face increased asthma attacks, frequent coughing, overnight hospitalization [18], and impaired lung functions, which increased the difficulty of treating and controlling asthma symptoms [19].

Different Factors are responsible for poor sleep quality including old age, female gender and lower level of education [9,20], uncontrolled asthma, severe asthma and longer duration of illness [9,15,18,20–22] poor social support, stress, poor sleep hygiene and substance use [20,23,24]).

The Global Initiative for Asthma (GINA) aims to improve asthma care by presenting evidence-based treatment options, but it does not consider the high prevalence of poor sleep quality in people with asthma. The treatment guideline does not include specific interventions given for poor sleep quality, nor instruct medical professionals on how to thoroughly record sleep histories on those individuals [1,25].

Although poor sleep quality is a common problem in people with asthma and has an impact on quality of life and work productivity, the magnitude of this problem and its determinant factors in those people are unknown because, to the best of the researcher's knowledge, no studies have been conducted in Ethiopia. Therefore, the objective of this study is to fill this gap and assess sleep quality and associated factors among people with asthma.

## Methods

### Study design and setting

An institutional based cross-sectional study design was conducted from June 06 to July 01/2022 on 406 people with asthma who had follow up at all public hospitals found in East Gojjam Zone. Its administrative city is Debre Markos, which is located around 300 km away from Addis Ababa, the capital city of Ethiopia, and 265 km away from Bahir Dar, the capital city of the Amhara region. According to the 2007 Ethiopian population census projection and East Gojjam Zone health department report, it has a total population of 2,153,937 and 506,520 households [26,27]. There are a total of eleven hospitals in this area. From these hospitals, there is only one comprehensive specialized hospital and one general hospital. The rest are primary hospitals. The hospitals include Debre Markos Comprehensive Specialized Hospital (DMCSH), Motta General Hospital (MGH), Bichena Primary Hospital (BPH), Shebel Berenta Primary Hospital (SBPH), Mertolemariam Primary Hospital (MLPH), Lumamie Primary Hospital (LPH), Debre Elias Primary Hospital (DEPH), Dejen Primary Hospital (DPH), Debre Werk Primary Hospital (DWPH), Yejubie Primary Hospital (YPH) and Bibugn Primary Hospital (BIPH).

### Population/ Study participants and sampling/ sample size estimations

**Source population.** People with asthma who have follow up at public hospitals found in East Gojjam Zone

**Study population.** People with asthma and who have follow-up at public hospitals found in East Gojjam Zone during the study period

**Eligibility criteria.**

**Inclusion criteria.** All people with asthma age 18 and older who have had follow-up for a minimum of two months were included in the study.

### Sample size determination and procedure

The sample size was determined by using the single population proportion formula with a 95% confidence interval level, a marginal error (d) of 5%, and a prevalence (P) of 60% [11] taken from a study that was conducted in Nigeria. The final sample size, including the 10% non-response rate, was 406. Participants were allocated proportionally from the 11 hospitals based on the number of patients visiting the chronic follow-up OPD of each hospital. The number of participants was estimated from the OPD registry of the chronic follow-up OPD of those hospitals. By considering the average data from January 09 to March 05, 2022, a total of 576 people with asthma had follow-up in the hospitals (188 in DMCSH, 57 in MGH, 47 in SBPH, 50 in BPH, 32 in MLPH, 39 in LPH, 34 in DPH, 20 in DEPH, 36 DWPH, 45 BiPH, and 28 in YPH). Proportionally, 133 individuals with asthma from DMCSH, 40 from MGH, 33 from SBPH, 35 from BPH, 23 from MLPH, 27 from LPH, 24 from DPH, 14 from DEPH, 25 from DWPH-32 from BiPH, and 20 from YPH were selected consequently after they finished their follow-up visit.

## Study variables

**Dependent variables.** Sleep quality (poor/good)

**Independent variables.** Socio-demographic variables (age, sex, marital status, residence, occupation and educational status), clinically related variables (level of asthma control, severity of asthma, duration of the illness and comorbidity), psychosocial related variables (social support and presence of comorbid common mental illness (stress and anxiety)) and behavioral related variables: sleep hygiene and substance use (alcohol, khat and cigarette)

## Operational definitions

**Poor sleep quality** explained by a cut-off point of greater than 5 by using Pittsburgh Sleep Quality Index (PSQI) score[28].

**Good sleep quality**: explained by a cut-off point of less than or equal to 5 by using Pittsburgh Sleep Quality Index (PSQI) score [28].

**Well controlled asthma** explained by a cut-off point of 20–25 by using asthma control test (ACT)[29].

**Not well controlled asthma** explained by a cut-off point of 16–19 by using ACT [29].

**Very poorly controlled asthma**: explained by a cut-off point of 5–15 by using ACT [29].

**Mild asthma** asthma that is well controlled with GINA Steps 1 or 2 treatment (low dose inhaled corticosteroid (ICS), with as needed short acting beta agonist (SABA) [1,25].

**Moderate asthma** is asthma that is well controlled with GINA step 3 treatment (low-dose inhaled corticosteroid/ long acting beta agonist (ICS/LABA) [1,25].

**Severe asthma**: is asthma that requires GINA Step 4/5 (moderate or high dose inhaled corticosteroid/ long acting beta agonist (ICS/LABA) plus add-on therapy), or remains uncontrolled despite this treatment [1,25].

**Poor social support** Explained by a cut-off point of 3–8 score based on Oslo Social Support Scale (OSSS) [30].

**Moderate social support** Explained by a cut-off point of 9–11 score based OSSS [30].

**Strong social support** Explained by a cut-off point of 12–14 score based on OSSS [30].

**Poor sleep hygiene** Explained by a cut-off point of greater than 16 based on SHI [31].

**Good sleep hygiene:** Explained by a cut-off point of less than or equal to16 based on SHI [31]

**Anxiety:** Explained by a cut-off point of greater than or equal to 8 based on DASS-21 [32]

**Stress:** Explained by a cut-off point of greater than or equal to 15 based DASS-21) [32]

**Current substance use** Use of at least anyone of substance (alcohol, khat, cigarette) in the past three months [33].

## Data collection tools and procedure

Data was collected by using pre-tested, structured, interviewer-administered questionnaires, which were prepared in English and translated into the local language (Amharic). The patients' charts were also reviewed to identify the drugs that the patients used.

Sleep quality was measured by the PSQI, which contains 19 items that are combined to form seven "component" scores (subjective sleep quality, sleep latency, sleep duration, habitual sleep efficiency, sleep disturbances, use of sleep medication, and daytime dysfunction), each of which has a range of 0–3 points. The seven component scores are then added to yield one "global" score with a range of 0–21 points, with "0" indicating no difficulty and "21" indicating severe difficulties in all areas [28]. The PSQI has good psychometric validity in Ethiopian adults, with sensitivity and specificity of 82 and 56.2%, respectively [34]. Its reliability in the current study was checked using the Cronbach $\alpha$, and it was 0.744.

The level of asthma control was measured by the ACT, which consists of five questions that assess activity limitation, shortness of breath, nighttime symptoms, use of rescue medication, and the patient's overall rating of asthma control

over the previous four weeks. The questions are scored from 1 (worst) to 5 (best), and the ACT score is the sum of the responses, giving a maximum best score of 25 [29]. The ACT is a reliable and valid tool with a sensitivity and specificity of 71% for detecting uncontrolled asthma [35].

The level of social support was assessed by the OSSS-3, which consisted of three items. The sum score ranges from 3 to 14, with high scores representing strong levels and low scores representing poor levels of social support. The tool is reliable and valid to measure social determinants of health in the general population [30,36].

Sleep hygiene was assessed by using SHI questionnaire comprised of 13 items. Each item is rated on a five-point scale ranging from 0 (never) to 4 (always). The total scores ranged from 0 to 52. Higher scores indicate poorer sleep hygiene status [31,37,38].

Anxiety and stress symptoms were measured by using the Depression Anxiety and Stress Scale (DASS 21), which consists of 21 questions, seven under each of the three negative affective states. Respondents indicate the extent to which they experienced each of the symptoms depicted in the items during the previous week on a 4-point Likert-type scale between 0 (did not apply to me at all) and 3 (applied to me very much, or most of the time). Finally the values obtained was multiplied by 2 [32,39]. DASS-21 is a reliable and suitable research tool useful for quick screening of anxiety and stress, as it was checked among Nigerian university students. The reliability of DASS-21 showed that it has excellent Cronbach's alpha values of 0.89 and 0.78 for the subscales of anxiety and stress, respectively [40].

Current uses of substances (alcohol, cigarettes, and khat) was assessed by identifying using those specific substances for nonmedical purposes in the last three months by using the Alcohol, Smoking and Substance Involvement Screening Test (ASSIST) [33].

The data was collected by 14 trained BSc nurses who were working out of the chronic disease follow-up outpatient department (OPD) and supervised by nurses who were BSc and above in each hospital for one month. After greeting the respondents and providing information about the purpose of the interview, the data collectors interviewed the participants and filled out the questionnaires while the participants were waiting to enter their follow-up OPD. The data collectors began the first interview with the participant who arrived first to the follow-up OPD. The questionnaires were pretested before the actual data collection on 5% of the sample size at Finote Selam General Hospital.

## Statistical analysis

The collected data were coded and entered in EPI-Data Version 4.4.2 and exported to SPSS-Version 25 Software-Package for Analysis. Frequency tables and graphs were used to describe the study variable. Bi-variable and multivariable logistic regression model was used to identify factors associated with poor sleep quality among people with asthma and to resolve confounding effect. In bi-variable logistic regression, variables with a P-value of <0.25 were transferred into multivariable logistic regression and finally variables which have P-value <0.05 in multivariate logistic regression were considered as statistically significant with dependent variable. The backward selection process was used to see the final associated variables. Model fitness was checked by Hosmer-Lemeshow goodness of fit test (p = 0.85) and interpreted as a model was fitted. Multicollinearity was checked using variance inflation factor (VIF) and its values were between 1.2–1.5, which was interpreted as no multicollinearity.

## Ethics approval and consent to participate

The study was approved by the ethical review board of the College of Health Sciences at Debre Markos University with a reference number of HSC/RC/Ser/PG/Co/212/11/14. The official letter was written to all hospitals found in East Gojjam Zone to get their permission. In addition, informed written consent was obtained from the respondent clients after explaining the purpose of the data collection. Confidentiality and privacy were maintained during data collection, analysis, and reporting; their names were not recorded.

## Results

### Socio demographic characteristics of the respondents

A total of 406 participants were included in this study, with a response rate of 100%. The median age of the participants was 40 (IQR = 24) years, with a range of 20–80 years, and 103 (25.4%) of them were within the age group of 30–39 years. About half of the participants, 209 (51.5%), were men, and 265 (65.3%) of them lived in urban areas. Concerning their marital status and occupation, 277 (68.2%) of the participants were married, and 129 (31.8%) were farmers, respectively. Regarding their educational status, 74 (18.2%) of them were unable to read and write (Table 1).

### Clinical characteristics of the participants

Only 113 (27.8%) of the participants had well-controlled asthma. About 141 (34.7%) had a comorbid illness, and 143 (35.2%) of them had asthma for more than 10 years. Regarding asthma severity, 121 (29.8%) of the participants had moderate asthma (Table 2).

Table 1. Socio-Demographic Characteristics of People with Asthma at East Gojjam Zone Public Hospitals, North-West Ethiopia, 2022 (N = 406).

| No | Variable | Category | Frequency | Percent (%) |
|---|---|---|---|---|
| 1 | Sex | Male | 209 | 51.5 |
| | | Female | 197 | 48.5 |
| 2 | Age group | 30-39 | 103 | 25.4 |
| | | 40-49 | 82 | 20.2 |
| | | 50-59 | 59 | 14.5 |
| | | >=60 | 63 | 15.5 |
| 3 | Marital status | Single | 64 | 15.8 |
| | | Married | 277 | 68.2 |
| | | Divorced | 30 | 7.4 |
| | | Widowed | 35 | 8.6 |
| 4 | Residence | Rural | 141 | 34.7 |
| | | Urban | 265 | 65.3 |
| 5 | Educational level | Unable to read and write | 74 | 18.2 |
| | | Read and Write | 157 | 38.7 |
| | | Primary school (1–8) | 26 | 6.4 |
| | | High school (9–12) | 38 | 9.4 |
| | | College/university student | 23 | 5.7 |
| | | Diploma | 36 | 8.9 |
| | | BSc and above | 52 | 12.8 |
| 6 | Occupation | Student | 33 | 8 |
| | | Farmer | 129 | 31.8 |
| | | Governmental | 79 | 19.5 |
| | | Daily laborer | 21 | 5.2 |
| | | Merchant | 43 | 10.6 |
| | | House wife | 68 | 16.7 |
| | | No job and Retired | 21 | 5.2 |
| | | Others | 12 | 3 |

NB: others: - security guard, carpenter, tailor

**Table 2. Clinical characteristics of people with asthma at East Gojjam Zone public hospitals, North-West Ethiopia, 2022 (N = 406).**

| No | Variables | Category | Frequency | Percent (%) |
|---|---|---|---|---|
| 1 | Level of asthma control | Very poorly controlled | 170 | 41.9 |
| | | not well controlled | 123 | 30.3 |
| | | well controlled | 113 | 27.8 |
| 2 | Severity of asthma | Severe asthma | 143 | 35.2 |
| | | Moderate asthma | 121 | 29.8 |
| | | Mild asthma | 142 | 35.0 |
| 3 | Duration of the illness | <=4 years | 140 | 34.5 |
| | | 5-10 years | 123 | 30.3 |
| | | >10 years | 143 | 35.2 |
| 4 | Presence of comorbidity | Yes | 141 | 34.7 |
| | | No | 265 | 65.3 |
| 5 | Types of comorbidity | Hypertension | 46 | 32.6 |
| | | Diabetes mellitus | 53 | 37.6 |
| | | kidney disease | 16 | 11.3 |
| | | Congestive heart failure | 20 | 14.2 |
| | | Other(epilepsy & arthritis) | 6 | 4.3 |

## Behavioral and psychosocial related characteristics

About 111 (27.3%) of the participants had poor social support, 247 (60.8%) had poor sleep hygiene behaviors, and 149 (36.7%) and 72 (17.7%) of them had anxiety and stress symptoms, respectively. About 261 (64.3%) participants use alcohol currently (table 3).

## Magnitude of poor sleep quality and the Pittsburgh Sleep Quality Index (PSQI) Subscale Scores

The overall prevalence of poor sleep quality was 64.5%, with a 95% CI of (60.1–69.2). About 100 (24.6%) of the study participants slept for more than 7 hours. Only 155 (38.2%) of the participants slept at least 85% of the time they spent in bed. The majority of the participants (285, or 70.2%) had disturbed sleep for less than one time a week. Almost all of the participants (402; 99%) did not use sleep medication, and 144 (35.5%) had daytime dysfunction three times a week (Table 4).

Regarding the subjective rating of sleep quality, only 72 (17.7%) rated their sleep quality as very good (Fig 1). In terms of their sleep latency, about 232 (57.1%) of the participants took 16–30 minutes to fall asleep (Fig 2).

## Factors associated with poor sleep quality

Age, marital status, asthma severity, the presence of comorbid diseases, the level of asthma control, social support, sleep hygiene, the presence of anxiety and stress symptoms, and current use of alcohol, khat, and cigarettes were all found to be associated with poor sleep quality in a bi-variable analysis at a p value less than 0.25.

In multivariable analysis, variables such as severe asthma, having comorbid diseases, very poorly controlled asthma, poor sleep hygiene, having anxiety symptoms, and current use of alcohol were statistically significant with poor sleep quality at a *p* value less than 0.05.

According to the result, people with severe asthma were 2.31 times more likely to have poor sleep quality when compared with people with mild asthma (AOR = 2.31, 95%CI = 1.17–4.59). Similarly, people with asthma who had chronic comorbid diseases were 3.11 times more likely to have poor sleep quality when compared with their counter parts (AOR = 3.11, 95%CI = 1.63–5.93).

**Table 3. The behavioral and psychosocial characteristics of people with asthma at East Gojjam Zone public hospitals, North-West Ethiopia, 2022 (N = 406).**

| No | Variables | Category | Frequency | Percent (%) |
|---|---|---|---|---|
| 1 | Social support | Poor | 111 | 27.3 |
| | | Moderate | 144 | 35.5 |
| | | Strong | 151 | 37.2 |
| | | No | 265 | 65.3 |
| 2 | Anxiety symptoms | Yes | 149 | 36.7 |
| | | No | 257 | 63.3 |
| 3 | Stress symptoms | Yes | 72 | 17.7 |
| | | No | 334 | 82.3 |
| 4 | Sleep hygiene | Poor | 247 | 60.8 |
| | | Good | 159 | 39.2 |
| 5 | Current alcohol use | Yes | 261 | 64.3 |
| | | No | 145 | 35.7 |
| 6 | Current cigarette use | Yes | 24 | 5.9 |
| | | No | 382 | 94.1 |
| 7 | Current khat use | Yes | 26 | 6.4 |
| | | No | 380 | 93.6 |

**Table 4. The Pittsburgh Sleep Quality Index (PSQI) Subscale Scores of People with Asthma at East Gojjam Zone Public Hospitals, North-West Ethiopia, 2022 (N = 406).**

| No | Component | Category | Frequency | Percent |
|---|---|---|---|---|
| 1 | Sleep duration (component three) | > 7 hours (0) | 100 | 24.6 |
| | | 6- 7 hours (1) | 208 | 51.2 |
| | | 5-6 hours (2) | 90 | 22.2 |
| | | < 5 hours (3) | 8 | 2.0 |
| 2 | Sleep efficiency (component four) | > 85% (0) | 155 | 38.2 |
| | | 75-84% (1) | 93 | 22.9 |
| | | 65-74% (2) | 104 | 25.6 |
| | | < 65% (3) | 54 | 13.3 |
| 3 | Sleep disturbance (component five) | Never (0) | 38 | 9.4 |
| | | 1 times a week (1) | 285 | 70.2 |
| | | 1-2 times a week (2) | 78 | 19.2 |
| | | >= 3 times a week (3) | 5 | 1.2 |
| 4 | Use of sleep medication (component six) | Never (0) | 402 | 99.0 |
| | | <1 time a week (1) | 3 | 0.7 |
| | | 1–2 times a week (2) | 1 | 0.2 |
| 5 | Day time dysfunction (component seven) | No problem (0) | 105 | 25.9 |
| | | 1-2 times a week (1) | 95 | 23.4 |
| | | 3 times a week (2) | 144 | 35.5 |
| | | >3 times a week (3) | 62 | 15.3 |
| 6 | Overall sleep quality | Poor | 262 | 64.5 |
| | | Good | 144 | 35.5 |

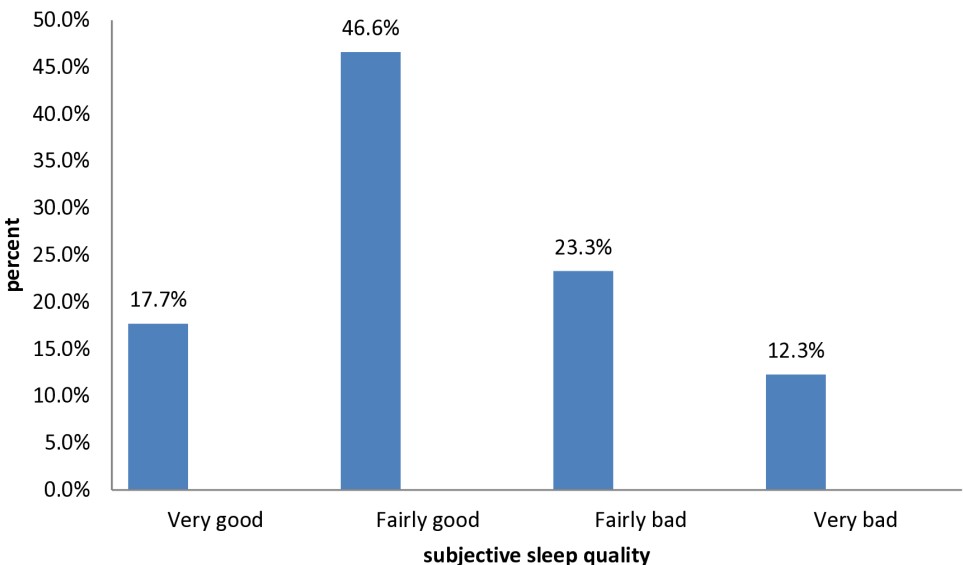

**Fig 1.  Rate of overall subjective sleep quality among people with asthma at East Gojjam Zone public hospitals, North-West Ethiopia, 2022 (n = 406).**

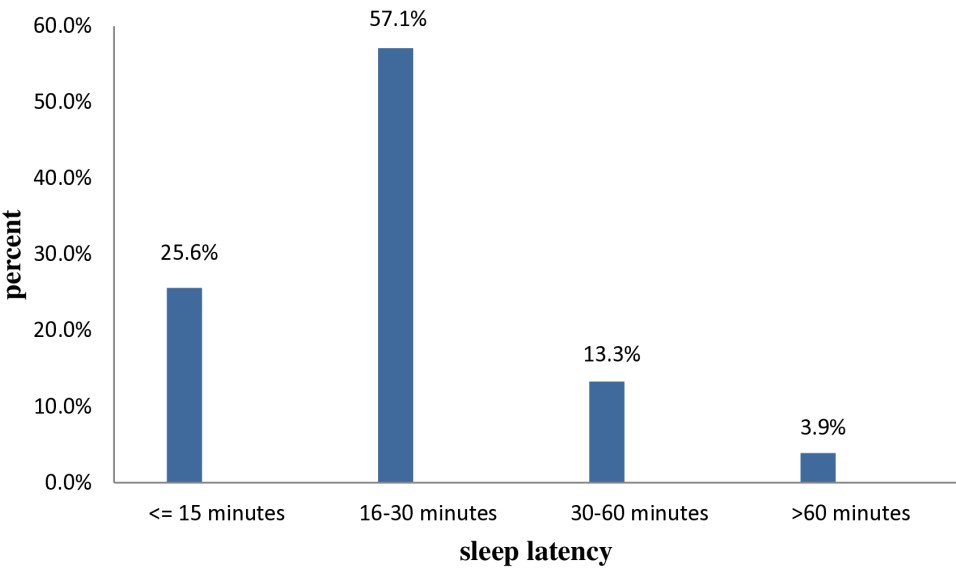

**Fig 2.  Sleep latency among people with asthma at East Gojjam Zone public hospitals, North-West Ethiopia, 2022 (n = 406).**

Regarding the level of asthma control, people with very poorly controlled asthma were 4.00 times more likely to be poor sleepers than people with well-controlled asthma (AOR = 4.00, 95% CI = 2.07–78.3). People with asthma who had poor sleep hygiene practices were 2.99 times more likely to have poor sleep quality than people who had good sleep hygiene practices (AOR = 2.99, 95% CI = 1.72–5.25).

People with asthma who had anxiety symptoms were 2.67 times more likely to have poor sleep quality than people without anxiety symptoms (AOR = 2.67, 95% CI = 1.43–5.00). Current alcohol use was also another factor associated with

**Table 5. Logistic regression showing association between factors and poor sleep quality among people with asthma at East Gojjam Zone public hospitals, North-West Ethiopia, 2022 (N = 406).**

| No | Variables | Category | Sleep quality | | COR(95%CI) | AOR (95%CI) |
|---|---|---|---|---|---|---|
| | | | Poor | Good | | |
| 1 | Age | 20-29 | 57 | 42 | 1 | 1 |
| | | 30-39 | 59 | 44 | .99(.57-1.73) | .75(.37-1.49) |
| | | 40-49 | 52 | 30 | 1.28(.7-2.33) | .93(.44-1.98) |
| | | 50-59 | 46 | 13 | 2.61(1.25-5.43) | 2.16(.88-5.27) |
| | | >=60 | 48 | 15 | 2.36(1.17-4.77) | 1.68(.71-3.99) |
| 2 | Marital status | Single | 34 | 30 | 1 | 1 |
| | | Married | 184 | 93 | 1.75(1.01-3.03) | 1.54(.69-3.39) |
| | | Divorced | 22 | 8 | 2.43(.94-6.25) | 1.57(.44-5.65) |
| | | Widowed | 22 | 13 | 1.49(.64-3.47) | .61(.18-2.12) |
| 3 | Asthma severity | Severe asthma | 114 | 29 | 2.88(1.69-4.87) | 2.31(1.17-4.59)* |
| | | Moderate asthma | 66 | 55 | .88(0.54-1.43) | .28(0.14-0.56) |
| | | Mild asthma | 82 | 60 | 1 | 1 |
| 4 | Comorbidity | Yes | 113 | 28 | 3.14(1.95-5.08) | 3.12(1.63-5.93)* |
| | | No | 149 | 116 | 1 | 1 |
| 5 | Level of asthma control | Very poorly controlled | 139 | 31 | 3.82(2.23-6.54) | 4.00(2.05-7.83)** |
| | | Not well controlled | 62 | 61 | .88(0.52-1.45) | .90(.49-1.65) |
| | | Well controlled | 61 | 52 | 1 | 1 |
| 6 | Social support | Poor | 87 | 24 | 2.26(1.29-3.95) | .78(.35- 1.76) |
| | | Moderate | 82 | 62 | .83(.52- 1.31) | .67(.35-1.28) |
| | | Strong | 93 | 58 | 1 | 1 |
| 7 | Sleep hygiene | Poor | 193 | 54 | 4.66(3.02-7.20) | 2.99(1.72-5.25)* |
| | | Good | 69 | 90 | 1 | 1 |
| 8 | Anxiety symptoms | Yes | 113 | 36 | 2.28(1.45-3.57) | 2.67(1.43-5.00)* |
| | | No | 149 | 108 | 1 | 1 |
| 9 | Stress symptoms | Yes | 54 | 18 | 1.82(1.02-3.24) | 2.06(1.00-4.28) |
| | | No | 208 | 126 | 1 | 1 |
| 10 | Current use of alcohol | Yes | 197 | 64 | 3.79(2.46-5.84) | 3.20(1.86-5.52)** |
| | | No | 65 | 80 | 1 | 1 |
| 11 | Current use of khat | Yes | 20 | 6 | 1.90(.75-4.85) | .54(.16-1.89) |
| | | No | 242 | 138 | 1 | 1 |
| 12 | Current use of cigarette | Yes | 20 | 4 | 2.89(.97-8.63) | 2.19(.47-10.31) |
| | | No | 242 | 140 | 1 | 1 |

NB: 1 = reference, AOR = adjusted odd ratio, COR = crude odd ratio,

* = p-value <0.05,

** = p-value<0.001

poor sleep quality. People with asthma who were current alcohol users were 3.20 times more likely to be poor sleepers when compared with people who were not current alcohol users (AOR = 3.20, 95% CI = 1.86–5.52) (Table 5).

## Discussion

Although poor sleep quality is a common issue for people with asthma and has an impact on quality of life and productivity at work, its prevalence and associated factors for those people were unknown because, as per the researchers'

knowledge, no research had been conducted in Ethiopia. The purpose of this study was to assess the prevalence of sleep quality and related factors among asthmatic individuals.

The findings of this study can help health care providers to develop better management plans that address both asthma symptoms and sleep disturbances, to improve sleep quality by adjusting medications or introducing sleep hygiene practices. The findings can also inform policy changes and updates to clinical guidelines, ensuring that the importance of sleep quality is recognized in the management of asthma.

In this study, the overall prevalence of poor sleep quality among people with asthma was found to be 64.5%, with a 95% CI of 60.1–69.2. The result was in line with the study done in Saudi Arabia (66%) [16].

However, the finding of this study was higher than the studies conducted in Owo and Ondo States in Nigeria (60%) [11], in Pakistan (54.5%) [10] and in Italy (58.3%) [15]. The discrepancy between the current study and studies conducted in Nigeria and Pakistan might be due to the smaller sample size in the studies conducted in Nigeria and Pakistan [41]. Another reason for the difference between the current study and the study conducted in Nigeria might be because individuals with asthma who had other comorbid chronic diseases were excluded from the study conducted in Nigeria [42]. The discrepancy between the current study and the study conducted in Italy might be due to the difference in socio-demographic characteristics of the participants between the two countries [43].

On the other hand the finding of this study was lower than the studies conducted in China (69.9%) [9], in Iran (72.4%) [14] in USA (79%) [13] and in Brazil (80.49%) [17]. This discrepancy might be due to socio-economic, psychosocial, and behavioral differences among the study participants [43].

In a multivariable logistic regression analysis, people with severe asthma were 2.31 times more likely to have poor sleep quality when compared with people with mild asthma. This was in line with the studies conducted in Italy [15], in Japan [21] and in London [44]. The symptoms of severe asthma are more intense, last for prolonged periods, and do not respond well to standard asthma treatments, which might be the reason why individuals with severe asthma were more likely to have poor sleep quality than individuals with mild asthma [45].

People with very poorly controlled asthma were 4.00 times more likely to be poor sleepers than individuals with well-controlled asthma. The result was consistent with the studies conducted in Italy [15], Japan [21], Spain [20], China [22] and USA [46]. This might be because individuals with very poorly controlled asthma have daily symptoms of asthma such as coughing, chest tightness, and shortness of breath and may experience nighttime flare-ups, which greatly affect their sleep quality [45].

People with asthma who had poor sleep hygiene practices were 2.99 times more likely to have poor sleep quality than their counterparts. The result was in line with a study done in the USA [23]. This could be due to poor sleep habits such as drinking stimulants (coffee) close to bedtime, having an irregular sleep-wake schedule that disrupts normal sleep patterns, infrequent exercise, watching television in bed, and reading in bed, all of which affect sleep quality [47].

People with asthma who had chronic comorbid diseases were 3.11 times more likely to have poor sleep quality when compared with people without other comorbid diseases. This might be due to the presence of other chronic comorbid diseases resulting in poor control of asthma symptoms, more complications, and increased emotional disorders, which might negatively affect the sleep quality of people with asthma [48].

People with asthma and current alcohol users were 3.20 times more likely to be poor sleepers than their counterparts. This might be because consuming alcohol leads to a number of nighttime awakenings due to snoring, and alcohol causes excessive relaxation of the throat muscles, which leads to narrowing or completely closing the airway, increasing resistance during inhalation, which directly and significantly impedes breathing and significantly and negatively affects sleep quality [49].

People with asthma who had anxiety symptoms were 2.67 times more likely to have poor sleep quality than their counterparts. This might be because people with anxiety have excess worry and fear, which makes it harder to fall asleep and stay asleep through the night. This can lead to sleep fragmentation, reducing both the quantity and quality of their sleep [50].

## Limitation of the study

The limitation of the study might be the nature of the study design, which was a cross-sectional study design that did not establish a temporal relationship between the outcome and the independent variables. Recall bias and social desirability bias might be the other limitations.

## Conclusion and recommendation

The magnitude of poor sleep quality among people with asthma was high in this study. Severe asthma, having comorbid diseases, very poorly controlled asthma, having poor sleep hygiene practices, having anxiety symptoms, and current alcohol use were significantly associated with poor sleep quality.

Since the magnitude of poor sleep quality is high, it needs great attention and remedial actions from health professionals, policymakers, the ministry of health, non-governmental organizations, and other concerned bodies to enhance the sleep quality of individuals with asthma. Reduction in alcohol consumption, improving sleep hygiene practices, and treating comorbidities would be effective measures for improving poor sleep quality. Giving special attentions for individuals with severe asthma and uncontrolled asthma symptoms is also needed. Individuals who participate in the preparation and revision of the treatment guideline should incorporate a sleep quality assessment check list into the guideline for people with asthma and put appropriate interventions in place for those who have poor sleep quality. It is better if future researchers conduct prospective follow-up studies in order to identify the cause-and-effect relationship between the factors and poor sleep quality.

## Supporting information

**S1 File: This S1 File is the date set used for this study.**
(RAR)

## Acknowledgments

The authors would like to thank the study participants for their cooperation and the staffs working in the hospitals found in East Gojjam Zone for their support during data collection

## Author contributions

**Conceptualization:** Menberu Gete, Bekalu Bewket, Balew Adane.

**Data curation:** Menberu Gete, Henok Mulugeta.

**Formal analysis:** Menberu Gete, Balew Adane, Dejen Tsegaye Alem.

**Funding acquisition:** Menberu Gete.

**Investigation:** Menberu Gete, Henok Mulugeta, Aster Tadesse.

**Methodology:** Menberu Gete, Tirusew Wondie, Aster Tadesse.

**Project administration:** Menberu Gete.

**Resources:** Menberu Gete.

**Software:** Menberu Gete, Tirusew Wondie, Bekalu Bewket, Dejen Tsegaye Alem, Tiliksew Liknaw Alemneh.

**Supervision:** Menberu Gete, Mezinew Sintayehu Bitew, Haile Amiha.

**Validation:** Menberu Gete, Wuhabie Tsega Sahilu, Tiliksew Liknaw Alemneh, Asmamaw Getnet.

**Visualization:** Menberu Gete, Baye Tsegaye Amlak.

**Writing – original draft:** Menberu Gete, Haile Amiha, Baye Tsegaye Amlak.

**Writing – review & editing:** Menberu Gete, Mezinew Sintayehu Bitew, Wuhabie Tsega Sahilu, Asmamaw Getnet.

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
