## [Decision Letter · Decision Letter 0]

26 Oct 2023

PONE-D-23-08092Sleep Quality and Associated Factors among People with Asthma at Public Hospitals in East Gojjam Zone, North West Ethiopia, 2022PLOS ONE

Dear Dr. Gete,

Thank you for submitting your manuscript to PLOS ONE. After careful consideration, we feel that it has merit but does not fully meet PLOS ONE’s publication criteria as it currently stands. Therefore, we invite you to submit a revised version of the manuscript that addresses the points raised during the review process.

 The Editor and reviewers’ comments are available towards the conclusion of this email, along with any suggestions from the Editorial Office regarding adjustments to formatting to adhere to the journal's policies.

We look forward to receiving your revised manuscript.

Kind regards,

Bereket Duko, PhD

Academic Editor

PLOS ONE

Journal Requirements:

3. PLOS requires an ORCID iD for the corresponding author in Editorial Manager on papers submitted after December 6th, 2016. Please ensure that you have an ORCID iD and that it is validated in Editorial Manager. To do this, go to ‘Update my Information’ (in the upper left-hand corner of the main menu), and click on the Fetch/Validate link next to the ORCID field. This will take you to the ORCID site and allow you to create a new iD or authenticate a pre-existing iD in Editorial Manager. Please see the following video for instructions on linking an ORCID iD to your Editorial Manager account: https://www.youtube.com/watch?v=_xcclfuvtxQ .

Additional Editor Comments (if provided):

Thank you for your nice work. While acknowledging the scientific validity of this study, I would like to kindly request that the authors make significant improvements to the manuscript to meet the quality standards required for the journal. The current version of the manuscript appears to have several shortcomings. I would strongly recommend that the authors seek support form a native English speaker for thorough editing. There are instances of spelling errors and incomplete sentences (grammatically incorrect) and formatting issues (e.g., table presentation) observed throughout the manuscript. I would suggest your attention to these details may greatly enhance the overall quality of the work. Authors also need to attend the following major points in addition to the reviewers' comments.

1. The sample size determination was based on a prevalence rate from a study conducted in Nigeria. To what extent can the prevalence of asthma and related factors be considered similar between Nigeria and East Gojjam Zone in Ethiopia, and how might this affect the generalizability of the findings? How might the regional and cultural differences between Nigeria and East Gojjam Zone affect the generalizability of the findings? I couldn’t find something related to this in the manuscript.

2. How did authors manage the potential sources of bias in data collection, given that the data was collected by trained nurses who were also working in the chronic disease follow-up outpatient department? Have you considered how might this affect the accuracy of the results?

3. The study considers various factors, including clinical, psychosocial, and behavioural variables, as potential predictors of poor sleep quality among people with asthma. Could you elaborate on the theoretical basis or prior research that supports the inclusion of these specific variables in the analysis?

4. This study was based on self-reported data for variables such as sleep quality, social support, and substance use. How might self-reporting introduce biases and limitations in the study's results, and what steps were taken to mitigate these potential issues?

5. Different data ascertainment tools were used to assess several factors. How does the reliability and validity of these assessment tools impact the robustness of your study’s findings, and were these tools validated for the specific population studied in East Gojjam Zone in Ethiopia?

6. Would you please provide evidence why did you use P-value of <0.25 to include variables in multivariable logistic regression (at least provide citations)?

7. The discussion section lacks a critical element. In addition to comparing your study with similar research, the authors should also delve into the significance of their findings within the context of Ethiopian or East African or African literature. Furthermore, it is essential for the authors to explore the policy, practice, and research implications arising from their findings.

Reviewers' comments:

Reviewer's Responses to Questions

**Comments to the Author**

1. Is the manuscript technically sound, and do the data support the conclusions?

Reviewer #1: Partly

Reviewer #2: Yes

2. Has the statistical analysis been performed appropriately and rigorously? 

Reviewer #1: Yes

Reviewer #2: Yes

3. Have the authors made all data underlying the findings in their manuscript fully available?

Reviewer #1: Yes

Reviewer #2: Yes

4. Is the manuscript presented in an intelligible fashion and written in standard English?

Reviewer #1: No

Reviewer #2: Yes

5. Review Comments to the Author

Reviewer #1: Comments on the manuscript entitled "Sleep Quality and Associated Factors among People with Asthma at Public Hospitals in East Gojjam Zone, North West Ethiopia, 2022"

Dear Editor, thank you very much for giving me the chance to read these manuscripts. The overall manuscript is done on important health problems, but I strongly recommend that authors modify the following issues before considering this article for publication:

Over all, the manuscript should be edited for English language and grammar issues.

Abstract: "The magnitude of poor sleep quality among people with... is widespread and has

detrimental consequences "This sentence is not full; it lacks words.

It is better to include the gap of this study in the background part of the abstract.

Introduction

It is better to show in detail the pathophysiology of sleep and asthma. I think, despite it not being done in the study area, there are studies that show related magnitude with almost related factors. What different considerations or approaches did researchers include for the current study? What is the difference between this study's different outputs of sleep quality and those of other medical illnesses, even those done in Ethiopia?

Methods

What is the reason behind using non-probability sampling? Is there any scientific reason to exclude participants with a follow-up of less than 2 months? I think it was also possible to take all 576 participants without considering allocation. What was the reason to use backward variable selection rather than the default enter method?

Result

Please limit the number of figures; it is not important to describe age separately by figure; rather, you can use sociodemographic tables. Similarly, for controlled asthma and the like,

18.2% of participants cannot read and write. Have you approached those regarding ethical issues for such people? Since nothing is reported for them.

From your findings, 64.3% of respondents use alcohol is currently. Which type of alcohol, and how could participants reported, specifies this question? I consider that almost all of the people taking medication for asthma are recommended to stop drinking, and your participants are also those with follow-ups lasting more than 2 months. How could 64.3% of participants use alcohol?

It is better to include all variables associated with bivariate logistic regression within the regression table rather than only those associated with multivariable analysis.

Discussion

The possible difference in findings should be well explained using scientific evidence, specifically how sample size and cultural variation make variation within the indicated countries, etc.

Limitation, please specify recall bias? For which data? How social desirability could be a limitation for this study

Your recommendation to the researcher: what is the reason for your being unable to do a follow-up study?

Reviewer #2: Comments

Thank you for coming up with this important topic. Overall, your study provides good insights into the sleep quality issues of people with Asthma.

Abstract

Page 2, line 23, you want to say with Asthma.

It would be worth it if you stated the gaps in the literature and then the aim in the introduction.

Recommendation: do you identify any gaps in the advice? a recommendation should be based on your findings. Should is not a good term for recommendations.

Introduction

The Introduction is good. But it can be excellent if you rewrite it to

For example, you can rewrite the first sentence like this:” Asthma is a chronic lung disease characterised by airway hyperresponsiveness to various direct and indirect stimuli, leading to variable symptoms of wheeze,,,,,,,,,,,,. Globally, it affects --- of individuals, accounting -% of individuals and responsible for, death …

Sleep is a crucial human physiological process important for the proper functioning of the brain and the overall well-being of the human body. It accounts for approximately an individual's life.

I would rewrite the whole introduction so that all the sentence structure and grammatical and punctuation errors will be resolved.

Method

The method part is very excellent and detailed, but it lacks synthesis.

-Delete and between MLPH & LPH

-I would detail the hospital settings, how many people they are serving, their care about Asthma

I would restructure the methods like this.

1. Study design and setting

2. Population/Study participants and sampling/ sample size estimations.

3. Study variables: you can state the variables and their definition simultaneously.

4. Data collection

5. Statistical analysis

Result

The results were well presented.

Discussion

I suggest adding a paragraph or two on the implications of your study in the discussion.

Limitation

How do you handle limitations such as recall and social desirability biases?

Conclusion

I would suggest saying “Conclusion and recommendation” or conclusion and stating the pertinent recommendations rather than having a separate section for recommendations.

6. PLOS authors have the option to publish the peer review history of their article (what does this mean? ). If published, this will include your full peer review and any attached files.

**Do you want your identity to be public for this peer review?** For information about this choice, including consent withdrawal, please see our Privacy Policy .

Reviewer #1: **Yes: ** Alemayehu Molla

Reviewer #2: No

---

## [Author Response · Author response to Decision Letter 1]

17 Jan 2024

Dear Editor, Reviewers,

Many thanks to you for your helpful feedback. We have now tried to address all your concerns to the best of our abilities. Kindly help us with your further concerns (if any) to improvise our paper.

---

## [Decision Letter · Decision Letter 1]

21 Oct 2024

PONE-D-23-08092R1Sleep Quality and Associated Factors among People with Asthma at Public Hospitals in East Gojjam Zone, North West Ethiopia, 2022PLOS ONE

Dear Dr. Gete,

Thank you for submitting your manuscript to PLOS ONE. After careful consideration, we feel that it has merit but does not fully meet PLOS ONE’s publication criteria as it currently stands. Therefore, we invite you to submit a revised version of the manuscript that addresses the points raised during the review process.

We look forward to receiving your revised manuscript.

Kind regards,

Zelalem Belayneh Muluneh

Academic Editor

PLOS ONE

Reviewers' comments:

Reviewer's Responses to Questions

**Comments to the Author**

1. If the authors have adequately addressed your comments raised in a previous round of review and you feel that this manuscript is now acceptable for publication, you may indicate that here to bypass the “Comments to the Author” section, enter your conflict of interest statement in the “Confidential to Editor” section, and submit your "Accept" recommendation.

Reviewer #3: (No Response)

Reviewer #4: All comments have been addressed

2. Is the manuscript technically sound, and do the data support the conclusions?

Reviewer #3: Partly

Reviewer #4: Yes

3. Has the statistical analysis been performed appropriately and rigorously? 

Reviewer #3: Yes

Reviewer #4: I Don't Know

4. Have the authors made all data underlying the findings in their manuscript fully available?

Reviewer #3: Yes

Reviewer #4: Yes

5. Is the manuscript presented in an intelligible fashion and written in standard English?

Reviewer #3: Yes

Reviewer #4: Yes

6. Review Comments to the Author

Reviewer #3: There are good efforts observed in the manuscript. I have given them some critical comments. If they address the issue, they will make it. Unless otherwise stated, they face challenges. So the manuscript might be accepted with major modifications.

Reviewer #4: Dear author(s), thank you for your manuscript.

The article has several notable strengths, including a clearly defined objective and a well-structured methodology. However, there are several flaws that weaken its effectiveness:

Specific comments follow:

1. One of the shortcomings of this study is the method of data collection, which is the interview.

First, explain who conducted the interview and then write the profile and background of the interviewer

Creating recall bias in this study: Data was collected using face-to-face interviews where participants had to recall their sleep patterns and asthma symptoms, which opens the study to recall bias. Participants may not accurately remember details; especially concerning sleep quality and other self-reported behaviors like alcohol consumption (What measures have you taken to prevent this type of bias?)

Creating social desirability bias in this study: Some participants may have underreported behaviors like alcohol use or overstated their sleep quality due to the social stigma around admitting poor health habits or conditions during face-to-face interviews (What measures have you taken to prevent this type of bias?)

2. Result: The results were well presented.

3. Discussion: In the discussion section, write a paragraph about the application of the findings in clinical settings.

7. PLOS authors have the option to publish the peer review history of their article (what does this mean? ). If published, this will include your full peer review and any attached files.

**Do you want your identity to be public for this peer review?** For information about this choice, including consent withdrawal, please see our Privacy Policy .

Reviewer #3: No

Reviewer #4: No

---

## [Author Response · Author response to Decision Letter 2]

30 Oct 2024

We appreciate you for your precious time in reviewing our paper and providing valuable comments. We have now tried to address all your concerns to the best of our abilities. Kindly help us with your further concerns (if any) to improvise our paper. The specific responses for each of your comments are highlighted in red.

---

## [Decision Letter · Decision Letter 2]

3 Jan 2025

PONE-D-23-08092R2Sleep Quality and Associated Factors among People with Asthma at Public Hospitals in East Gojjam Zone, North West Ethiopia, 2022PLOS ONE

Dear Dr. Gete,

Thank you for submitting your manuscript to PLOS ONE. After careful consideration, we feel that it has merit but does not fully meet PLOS ONE’s publication criteria as it currently stands. Therefore, we invite you to submit a revised version of the manuscript that addresses the points raised during the review process.

**A**

We look forward to receiving your revised manuscript.

Kind regards,

Zelalem Belayneh Muluneh

Academic Editor

PLOS ONE

Reviewers' comments:

Reviewer's Responses to Questions

**Comments to the Author**

Reviewer #4: All comments have been addressed

Reviewer #5: (No Response)

2. Is the manuscript technically sound, and do the data support the conclusions?

Reviewer #4: Yes

Reviewer #5: Yes

3. Has the statistical analysis been performed appropriately and rigorously? 

Reviewer #4: Yes

Reviewer #5: Yes

4. Have the authors made all data underlying the findings in their manuscript fully available?

Reviewer #4: Yes

Reviewer #5: Yes

5. Is the manuscript presented in an intelligible fashion and written in standard English?

Reviewer #4: Yes

Reviewer #5: Yes

6. Review Comments to the Author

Reviewer #4: (No Response)

Reviewer #5: Poor sleep quality is well known among asthmatic patients and it is worst in the poorly controlled asthma group due to the frequent nocturnal symptoms that the patients have. The prevalence of poor sleep quality in the paper was acquired by subjective feeling of the patients (self rated - figure 1). I can't find the prevalence based on the PSQI scorer the overall prevalence of poor sleep quality in the study population in this paper.

This study should also exclude OSA patients and patients with GERD symptoms which are known to have sleep disturbances.

7. PLOS authors have the option to publish the peer review history of their article (what does this mean? ). If published, this will include your full peer review and any attached files.

**Do you want your identity to be public for this peer review?** For information about this choice, including consent withdrawal, please see our Privacy Policy .

Reviewer #4: No

Reviewer #5: No

---

## [Author Response · Author response to Decision Letter 3]

21 Jan 2025

Many thanks to you for your helpful feedback. We have now tried to address all your concerns to the best of our abilities. Kindly help us with your further concerns (if any) to improvise our paper.

---

## [Decision Letter · Decision Letter 3]

21 Mar 2025

PONE-D-23-08092R3Sleep Quality and Associated Factors among People with Asthma at Public Hospitals in East Gojjam Zone, North West Ethiopia, 2022PLOS ONE

Dear Dr. Gete,

Thank you for submitting your manuscript to PLOS ONE. After careful consideration, we feel that it has merit but does not fully meet PLOS ONE’s publication criteria as it currently stands. Therefore, we invite you to submit a revised version of the manuscript that addresses the points raised during the review process.

We look forward to receiving your revised manuscript.

Kind regards,

Zelalem Belayneh

Academic Editor

PLOS ONE

Journal Requirements:

Reviewers' comments:

Reviewer's Responses to Questions

**Comments to the Author**

1. If the authors have adequately addressed your comments raised in a previous round of review and you feel that this manuscript is now acceptable for publication, you may indicate that here to bypass the “Comments to the Author” section, enter your conflict of interest statement in the “Confidential to Editor” section, and submit your "Accept" recommendation.

Reviewer #5: All comments have been addressed

2. Is the manuscript technically sound, and do the data support the conclusions?

Reviewer #5: Yes

3. Has the statistical analysis been performed appropriately and rigorously? 

Reviewer #5: Yes

4. Have the authors made all data underlying the findings in their manuscript fully available?

Reviewer #5: Yes

5. Is the manuscript presented in an intelligible fashion and written in standard English?

Reviewer #5: Yes

6. Review Comments to the Author

Reviewer #5: (No Response)

7. PLOS authors have the option to publish the peer review history of their article (what does this mean? ). If published, this will include your full peer review and any attached files.

**Do you want your identity to be public for this peer review?** For information about this choice, including consent withdrawal, please see our Privacy Policy .

Reviewer #5: No

---

## [Author Response · Author response to Decision Letter 4]

25 Mar 2025

Thank you, Dear Editor for your feedback regarding our reference list. As requested, we have carefully reviewed all cited references to ensure accuracy and completeness. We have also conducted a thorough check for any retracted articles and confirm that none of the references included in our manuscript have been retracted. Since no retracted papers are cited, no changes have been made to our reference list.

---

## [Editor Report · Decision Letter 4]

6 Apr 2025

Sleep Quality and Associated Factors among People with Asthma at Public Hospitals in East Gojjam Zone, North West Ethiopia, 2022

PONE-D-23-08092R4

Dear Dr. Menberu Gete,

We’re pleased to inform you that your manuscript has been judged scientifically suitable for publication and will be formally accepted for publication once it meets all outstanding technical requirements.

Kind regards,

Zelalem Belayneh

Academic Editor

PLOS ONE
---

## [Editor Report · Acceptance letter]

PONE-D-23-08092R4

PLOS ONE

Dear Dr. Gete,

I'm pleased to inform you that your manuscript has been deemed suitable for publication in PLOS ONE. Congratulations! Your manuscript is now being handed over to our production team.

Kind regards,

on behalf of

Mr. Zelalem Belayneh

Academic Editor

PLOS ONE